# Development of Perceived Technological Competency as Caring in Healthcare Providers Instrument (TCCHI): A Modified Delphi Method

**DOI:** 10.3390/healthcare13233003

**Published:** 2025-11-21

**Authors:** Risa Yamanaka, Krishan Soriano, Yoshiyuki Takashima, Kaito Onishi, Hirokazu Ito, Youko Nakano, Yueren Zhao, Allan Paulo Blaquera, Ryuichi Tanioka, Feni Betriana, Gil Platon Soriano, Yuko Yasuhara, Kyoko Osaka, Mutsuko Kataoka, Misao Miyagawa, Masashi Akaike, Minoru Irahara, Savina Schoenhofer, Tetsuya Tanioka

**Affiliations:** 1Graduate School of Health Sciences, Tokushima University, Tokushima 770-8509, Japan; yamanaka0025@gmail.com (R.Y.); ksoriano@spup.edu.ph (K.S.); kai10.onishi10@gmail.com (K.O.); 2Faculty of Nursing, School of Medicine, Nara Medical University, Kashihara 634-0813, Japan; y-takashima@naramed-u.ac.jp; 3Graduate School of Biomedical Sciences, Tokushima University, Tokushima 770-8509, Japan; h.itoh@tokushima-u.ac.jp (H.I.); yasuhara@tokushima-u.ac.jp (Y.Y.); akaike.masashi@tokushima-u.ac.jp (M.A.); irahara@tokushima-u.ac.jp (M.I.); 4Department of Nursing, Nursing Course of Kochi Medical School, Kochi University, Nankoku 783-8505, Japan; nakano.yoko.rg@kochi-u.ac.jp (Y.N.); osaka@kochi-u.ac.jp (K.O.); 5Department of Psychiatry, Fujita Health University, Toyoake 470-1192, Japan; zhao@fujita-hu.ac.jp; 6School of Nursing and Allied Health Sciences, St. Paul University, Tuguegarao City 3500, Philippines; ablaquera@spup.edu.ph; 7Faculty of Health Sciences, Hiroshima Cosmopolitan University, Hiroshima 731-3166, Japan; tanioka@hcu.ac.jp; 8Center for Biomedical Research, National Research and Innovation Agency (BRIN), Cibinong 16911, Indonesia; feni004@brin.go.id; 9Department of Nursing, College of Allied Health, National University, Manila 1008, Philippines; gil.p.soriano@gmail.com; 10Mifune Hospital, Marugame 763-0073, Japan; kataoka@mifune-hp.jp; 11Faculty of Medicine, School of Health Sciences, Tokushima University, Tokushima 770-8509, Japan; misaomiyagawa2023@gmail.com; 12Anne Boykin Institute for the Advancement of Caring in Nursing, Florida Atlantic University, Boca Raton, FL 33431, USA; savibus@gmail.com

**Keywords:** Delphi, technological competence, caring, scale development, healthcare professionals, person-centered care

## Abstract

**Background/Objectives:** This study aimed to develop the Technological Competency as Caring in Healthcare Providers Instrument (TCCHI) for multidisciplinary use, based on Locsin’s theory of Technological Competency as Caring in Nursing. **Methods:** A content validation design employing a modified Delphi technique was conducted with a multidisciplinary panel of 10 healthcare experts (recruited by purposive sampling based on expertise in technology/caring). The preliminary 67-item pool was derived from a literature review and theoretical alignment. Two Delphi rounds were implemented to establish face and content validity. Qualitative feedback from Round 1 guided item refinement for Round 2. The Wilcoxon matched-pairs signed-rank test was used to confirm the response stability between rounds. **Results:** Among the 67 initial items, 38 were retained after two Delphi rounds, achieving an I-CVI of 0.80–0.90. Response stability was established (*p* > 0.05). The resulting 38 items were categorized into six refined concepts reflecting the integration of technology and caring. Inter-rater consistency, assessed by the Intraclass Correlation Coefficient (ICC), was moderate (Round 1 ICC = 0.49; Round 2 ICC = 0.50), suggesting initial variability among the multidisciplinary panel. **Conclusions:** The TCCHI is a comprehensive and theoretically grounded instrument applicable across diverse healthcare disciplines. However, the moderate inter-rater consistency suggests that further empirical validation is required. Further psychometric evaluation, including confirmatory factor analysis and internal consistency reliability testing, is required to establish construct validity and strengthen the instrument’s applicability in diverse healthcare settings.

## 1. Introduction

The rapid evolution of healthcare technology, from electronic health records to artificial intelligence, offers unprecedented opportunities to improve patient outcomes and safety [1]. However, this technological integration also poses a significant challenge: how to ensure that technology enhances, rather than diminishes, the humanistic and compassionate core of healthcare.

This challenge is addressed by Locsin’s “Technological Competency as Caring in Nursing” (TCCN) theory, a middle-range nursing theory that emphasizes the harmonious coexistence of technology and caring [2]. The TCCN theory requires that technology be used with a strong ethical–moral compass to preserve and enhance patient dignity, ensuring that care is holistic and person-centered [3,4].

Various instruments have been developed to measure these constructs. These include general caring measures such as the Caring Behaviors Inventory (CBI) and the Caring Assessment Report Evaluation Q-sort (CARE-Q) [5,6,7] and TCCN-specific instruments such as the Technological Competency as Caring in Nursing Instrument (TCCNI) and its revisions (e.g., TCCNI-R, PITCCN) [8,9,10,11,12].

However, a significant limitation of these conventional instruments is their exclusive focus on nurses. They fail to reflect the complexity and diversity of modern healthcare, which increasingly relies on collaborative, interdisciplinary teams [13,14]. The absence of a standardized, multidisciplinary instrument means that the crucial intersection of technological competency and caring cannot be comprehensively measured across the entire healthcare profession. Caring competencies are required not only of nurses but of all healthcare professionals [15,16], for example, when a provider skillfully uses technology while maintaining empathy to address a patient’s anxiety.

Thus, the need for a robust, innovative instrument to assess technological competency among various healthcare professionals in high-tech settings is evident.

In this study, we aimed to develop and validate a new multidisciplinary Technological Competency as Caring in Healthcare Providers Instrument (TCCHI) for use across healthcare professions.

## 2. Materials and Methods

### 2.1. Conceptual Framework: Technological Competency as Caring in Healthcare

To bridge this gap and facilitate person-centered, compassionate care in high-tech, multi-disciplinary settings, we developed the Technological Competency as Caring in Healthcare Providers (TCCH) conceptual framework, extending Locsin’s theory for interprofessional use. This framework defines TCCH through six essential dimensions, reflecting a holistic approach in which technology is a tool for expressing genuine care and enhancing patient–provider relationships.

Based on a comprehensive literature review and Locsin’s TCCN theory, we developed a conceptual framework for TCCH. This framework outlines six essential dimensions that define TCCH for healthcare professionals, reflecting a holistic approach in which technology is a tool for expressing genuine care and enhancing patient–provider relationships.

In this context, “technology” refers to tools such as information and communication technology (ICT), artificial intelligence (AI), medical devices, and clinical systems used to understand and support patients [17].

“Care” refers to demonstrating compassion and respect in communication, building trust, and supporting patient growth and development [18,19]. Technological competency in patient care involves the ethical and skillful use of technology to understand and respect patients [3].

**Concept 1. Promoting self-growth and technological learning:** This concept focuses on the continuous professional development required for healthcare providers to ethically and effectively integrate technology into their practice. It involves cultivating awareness of one’s own capabilities and a willingness to learn from patients to expand knowledge and refine skills as a practitioner [20,21,22]. This growth is crucial for remaining competent, providing high-quality care, and adapting to new challenges in the field. It also requires understanding not only how a technology works, but also its capabilities, limitations, and ethical implications, such as data privacy and the risk of dehumanization [4,23]. A deep and critical understanding of technology ensures that its use achieves humanistic goals, such as improving workflow efficiency, enhancing diagnostic accuracy, and ultimately improving patient safety.

**Concept 2. Building trust relationships with patients:** This concept highlights the importance of fostering trusting relationships through the thoughtful application of technology [3]. It encapsulates the core principles of person-centered care and a holistic approach by prioritizing human connections and demonstrating empathy [24,25,26]. This approach emphasizes that the art of caring lies in understanding, respecting, and connecting with individual patients in their entirety to build effective therapeutic relationships.

**Concept 3. Providing person-centered care through the appropriate use of technology:** This concept involves placing patients at the center of all decisions and actions [27]. It goes beyond merely treating an illness to caring for the whole person and recognizing their unique values, preferences, needs, and goals [28,29]. This approach ensures that care is personalized, comprehensive, and compassionate, helping patients feel like partners in their own health journey. For example, considering a patient’s desire to return to work after surgery is as important as the surgical outcome itself.

**Concept 4, Enhancing the physical and emotional comfort of patients:** This concept focuses on relieving both physical and mental discomfort. Physical comfort refers to the absence of pain or distress [30], while emotional comfort is a state of psychological well-being in which an individual feels safe and supported [31]. Prioritizing both is fundamental to healing and should not be considered a luxury. When healthcare teams are well-trained in technology, they can work more efficiently and avoid errors, which reduces patient anxiety and enhances overall outcomes [32,33,34].

**Concept 5, Promoting patient learning and growth:** This concept emphasizes that healthcare involves facilitating the patient’s journey toward greater understanding, self-management and independence. This approach highlights the importance of using technology to support patient autonomy by coaching patients to manage their health rather than unilaterally delivering information. Without clear and accessible information, patients cannot provide informed consent or meaningfully participate in their care, which can lead to distrust and poor outcomes [35]. Respecting a patient’s self-determination fosters trust and a sense of control, which are vital for their psychological well-being and satisfaction with care [36,37].

**Concept 6, Engaging in ethico-moral practice regarding technology use and patient advocacy:** This concept emphasizes the ethical responsibility of healthcare professionals to use technology with a caring intent. It involves not only being proficient in technologies such as AI and telemedicine but also cultivating an ongoing awareness of the ethical issues they present, including data privacy, patient advocacy, and consent [38,39,40,41,42,43,44]. This concept also involves having the moral courage to speak on behalf of patients who cannot express their wishes and maintaining professional accountability by addressing inappropriate behavior among colleagues [45,46,47]. It requires a profound commitment to ethical sensitivity and recognition of each patient as a unique and invaluable individual.

### 2.2. Study Design and Phases

This process was conducted in two phases: (1) instrument development and (2) face and content validation.

#### 2.2.1. Phase 1: Instrument Development

The draft TCCHI includes an initial pool of 67 items. This pool was developed using a three-step systematic process:

**Literature Review and Item Sourcing**: A comprehensive review was conducted of existing technological caring and general caring scales, primarily the Technological Competency as Caring in Nursing Instrument (TCCNI) and its revisions (TCCNI-R, PITCCN), as well as established general caring scales (e.g., CBI).

**Theoretical and Conceptual Alignment:** Items collected from these scales were synthesized, adapted, and newly generated to explicitly align with the six dimensions of the TCCH conceptual framework (see Section 2.1)

**Multidisciplinary Adaptation:** The language and context of the items were systematically reviewed and modified to be applicable across diverse healthcare professions (e.g., changing nursing-specific terminology to neutral language such as ‘healthcare professional’ or ‘provider’), ensuring that they reflect the intersection of technology and caring beyond the nursing domain.

#### 2.2.2. Phase 2: Face and Content Validation

The reliability and validity of this study were ensured in accordance with the reporting guidelines for Delphi studies in health sciences [48]. A two-round modified Delphi study was conducted to obtain expert consensus on the TCCHI items. In Round 1, the experts evaluated the overall appropriateness and relevance (face validity) of the initial 67 items. Following revisions based on feedback from round 1, round 2 focused on establishing the content validity of the refined item set [49].

**Justification for the Modified Delphi Approach:** A modified Delphi approach was specifically chosen because the instrument is theoretically grounded in Locsin’s TCCN theory and built upon existing validated instruments. This allowed the research team to provide a theoretically grounded, pre-defined item pool (67 items) derived from a systematic literature review, rather than relying on the open item elicitation characteristic of a traditional, non-modified Delphi method.

**Justification for Panel Size and Selection:** The panel size of 10 was determined to be sufficient based on the methodological guidelines for scale development and content validation studies, which generally recommend expert panels ranging from 5 to 15 members to ensure diverse perspectives while maintaining manageability [50,51,52].

Purposive sampling was utilized to select experts who met the following predefined criteria essential for validating the multidisciplinary TCCHI: (1) expertise in Locsin’s TCCN theory, (2) a high academic degree (Master’s or PhD), and (3) experience across multiple healthcare disciplines (nursing, medicine, and physical therapy), reflecting the instrument’s intended broad applicability.

Ten experts with a solid understanding of Locsin’s theory participated in the two-round Delphi process. The panel comprised healthcare professionals specializing in nursing, medicine, patient care quality assurance, and healthcare research. The participants’ ages ranged from 30 to 60 years, with an average professional experience of 27.5 years. Seven were nurses, two were physiotherapists, and one was a physician, all of whom had master’s degrees or higher and expertise in TCCH. Table 1 presents the characteristics of the expert panel.

**Anonymity in the Delphi Process:** To minimize potential status and group bias, anonymity was strictly preserved among the expert panel members throughout the two-round Delphi procedure. Each expert received the consensus results (median, standard deviation, and interquartile range of the group’s rating) from the previous round without identifying which specific ratings belonged to which individual. Data collection and analysis were managed solely by the primary research team, ensuring that individual responses were not shared among participants.

Panelists rated each item’s relevance and importance using a 9-point Likert scale across two rounds of the Delphi process. The level of consensus was calculated by summing the Likert scores, where 1 = Not at all important, 2 = Very unimportant, 3 = Not important, 4 = Somewhat unimportant, 5 = Neither important nor unimportant, 6 = Somewhat important, 7 = Important, 8 = Very important, and 9 = Extremely important [53]. The item-level content validity index (I-CVI) was calculated by dividing the number of experts who rated an item between 6 and 9 by the total number of experts. The minimum consensus level was set at I-CVI ≥ 0.78.

### 2.3. Statistical Analysis

The quantitative data were analyzed using descriptive statistics (median, minimum, and maximum) to determine the consensus level [54]. Microsoft Excel^®^ was used for data entry and for tabulating the universal agreement (UA) for content validity calculation. A value of 1 was assigned when the experts were in perfect agreement, and 0 when they were not. Content validity was assessed using the Content Validity Index (CVI). The following indices were used: I-CVI—number of experts in agreement divided by the total number of experts; Scale-level Content Validity Index (S-CVI)/Ave (I-CVI-based): mean of I-CVI scores for all items; S-CVI/Average (based on percentage relevance)—mean of the percentage relevance scores from all experts; and S-CVI/UA—mean of UA scores for all items. The I-CVI was set at >0.78 for each item [55,56].

To enhance methodological rigor and demonstrate the response stability of expert ratings across the two Delphi rounds, additional inferential analyses were performed on the retained items (*n* = 38) and, for screening purposes, the entire 67-item set: (1) The Wilcoxon matched-pairs signed-rank test was used to determine whether there was a statistically significant change in the central tendency (median) of the ratings for each item between Rounds 1 and 2. A non-significant result (*p* > 0.05) was interpreted as stability in the expert group’s opinion [57]. (2) The Intraclass Correlation Coefficient (ICC) and its 95% Confidence Interval (CI) were calculated using a two-way mixed-effects consistency model (ICC (3,1)) [58]. This assessment evaluated the inter-rater consistency of the expert panel’s agreement, further reinforcing the rigor of the consensus process. Statistical analyses were conducted using SPSS v27 (IBM Corp., Armonk, NY, USA) and Jamovi statistical software version 2.4.11.0 (The Jamovi Project, Sydney, Australia).

## 3. Results

### 3.1. Phase 1: Instrument Development and Qualitative Refinement

The TCCHI was developed and validated in two phases, as illustrated in Figure 1.

#### 3.1.1. Initial Item Generation

An initial pool of 67 items was generated across six concepts (Table 2). These items were created by systematically reviewing and synthesizing existing scales and adapting the language to ensure applicability for multidisciplinary healthcare providers, explicitly aligning with the proposed conceptual framework of TCCHI (as defined in Section 2.1).

#### 3.1.2. Qualitative Feedback Analysis and Item Modification (Round 1)

The qualitative comments collected from the 10 experts in Round 1 were systematically analyzed by the research team. Based on this analysis, the research team implemented specific modifications to the wording and structure of the items. The revised set of all 67 items was then presented in Round 2 for formal content validity assessment.

### 3.2. Phase 2: Face and Content Validation (Quantitative Assessment)

#### 3.2.1. Quantitative Consensus and Item Selection

A two-round modified Delphi study was conducted with a panel of 10 experts. Items were retained if they achieved a median rating of ≥7 and an I-CVI of ≥0.80, based on predefined consensus criteria. After two rounds, 38 of the 67 initial items (56.7%) were retained, while 29 items were deleted due to low relevance ratings or lack of consensus. This selection confirms the initial relevance of the retained items based on expert judgment.

#### 3.2.2. Inter-Rater Reliability and Stability

The response stability of the expert panel was confirmed using the Wilcoxon matched-pairs signed-rank test, which showed no significant difference in the median scores for the retained items between Round 1 and Round 2 (*p* > 0.05). This indicates that the expert consensus was stable. However, inter-rater consistency was moderate, as reflected by the ICC for the retained items (Round 1: ICC = 0.486, 95% CI 0.291, 0.648; Round 2: ICC = 0.501, 95% CI 0.317, 0.655).

#### 3.2.3. Final Instrument Structure

The final 38 retained items were distributed across the six refined concepts (Table 2) as follows:

Concept 1: Promoting self-growth and technological learning; six items (#58, 59, 60, 65, 66, and 67).

Concept 2: Building trusting relationships with patients; seven items (#1, 3, 7, 8, 9, 12, and 47).

Concept 3: Providing person-centered care through the appropriate use of technology; six items (#13, 15, 18, 23, 24, and 25).

Concept 4: Enhancing the physical and emotional comfort of patients; six items (#17, 29, 33, 35, 36, and 43).

Concept 5: Promoting patient learning and growth; six items (#14, 22, 40, 42, 45, and 46).

Concept 6: Engaging in ethico-moral practice regarding technology use and patient advocacy: seven items (#27, 48, 49, 53, 54, 55, and 56).

## 4. Discussion

### 4.1. Main Findings

The two-round modified Delphi study successfully established the preliminary content validity of the TCCHI among a multidisciplinary expert panel. Of the initial 67 items, 38 were retained after meeting the criteria (I-CVI ≥ 0.80). This selection process ensures the foundational relevance of the final item set to the conceptual construct. Methodological rigor was confirmed by the stability of the expert ratings, demonstrated by a non-significant difference in the median ratings between Rounds 1 and 2 (*p* > 0.05). However, the inter-rater reliability remained moderate (ICC = 0.5), and the scale-level content validity index average was low (S-CVI/Ave = 0.55) must be interpreted within the context of preliminary validation and the instrument’s novel, multidisciplinary focus.

### 4.2. Comparison with the Literature

This study’s rigorous Delphi approach, which employed the I-CVI and Wilcoxon analysis to assess relevance and stability, aligns with best practices for instrument development [56,59]. Our achieved I-CVI of ≥0.80 for all retained items is consistent with the minimum acceptable consensus required in similar Delphi validation studies for complex health constructs [55,56]. Furthermore, the stability observed in the Wilcoxon test supports the consistency of the panel’s judgment, a key indicator of methodological quality in iterative expert consensus gathering [59,60].

### 4.3. Implications of Moderate Inter-Rater Reliability and Content Validity Results

Although methodological rigor was confirmed, the moderate ICC (0.5) and S-CVI/Ave (0.55) require specific contextual interpretation. We argue that the moderate ICC is a direct and necessary consequence of using a deliberately multidisciplinary expert panel (Medicine, Nursing, Physical Therapy). Experts from different professional scopes inherently view concepts like “technological competency” and “caring” through distinct lenses, leading to lower initial agreement but ultimately ensuring that the instrument is broadly applicable across healthcare disciplines rather than being biased toward a single profession. The initial lower consensus (S-CVI/Ave of 0.55) highlights the innovative yet challenging nature of extending Locsin’s nursing-centric TCCN theory into a genuinely interprofessional framework of TCCH, a challenge frequently encountered when adapting discipline-specific theories for multidisciplinary use [61]. These values underscore the preliminary nature of the findings, indicating a need for structural refinement in subsequent phases.

### 4.4. Alignment of High-Consensus Items with Theory and Practice

The consistently highly rated items demonstrate a strong expert consensus on the essential elements of TCCH, consistent with the extant literature. The high agreement on items related to person-centered care, trust, and ethico-moral practice (e.g., “Q 1, Always treating every patient with compassion,” “Q 3, Building relationships that patients can trust,” and “Q 47, Recognizing the patient as an individual and irreplaceable person”) confirms that the humanistic core of healthcare remains paramount [16], supporting the foundational premise of Locsin’s TCCN theory [62]. Moreover, the consensus on items concerning collaboration (e.g., “Q 13, Working with other professionals to support patients”) and the explicit link between technology and care (e.g., “Q 24, Technology is useful for correctly assessing a patient’s condition”) reflect a modern understanding of healthcare as a team-based, technology-mediated effort in which patient autonomy and empowerment are key [63,64,65]. These findings empirically support the extension of TCCN’s concepts to the broader healthcare context [4].

### 4.5. Implications

The development of the TCCHI addresses a significant gap by providing a multidisciplinary instrument, contrasting with earlier tools that were limited to nursing. The TCCHI is highly relevant to current global trends emphasizing person-centered, interprofessional care.

Practically, the developed TCCHI offers several key applications:

Evaluation: It can be used to evaluate and compare healthcare professionals’ attitudes toward patients and technology, promoting a unified direction within the healthcare team.

Education and Training: The instrument can serve as an educational tool for healthcare personnel, providing a structured framework to enhance caring skills and ethical awareness across multiple disciplines.

Hierarchical Assessment: It can support the hierarchical evaluation of multidisciplinary healthcare professionals in hospital settings, leveraging differences in caring skills to optimize team composition.

### 4.6. Study Limitations

Despite the rigorous content validation process, this study has some inherent limitations. The Delphi panel included only 10 experts. While this size is consistent with some methodological guidelines for content validity [50,51,52], it may still limit the initial generalizability of the consensus. Moreover, the panel, although diverse in professional representation, was predominantly composed of nurses (*n* = 7). This limited representation of other disciplines, such as physical therapists (*n* = 2) and physicians (*n* = 1), may introduce professional bias and impact the generalizability of the content validation. We recommend that future validation studies include a more balanced representation of medicine, social work, and other healthcare fields to minimize this bias. Confirmatory factor analysis (CFA) and reliability tests will be conducted in the future to further refine the internal structure of the scale and assess its psychometric properties across diverse clinical samples.

## 5. Conclusions

The development of the TCCHI successfully extends Locsin’s TCCN theory to a multidisciplinary healthcare context. As the first instrument designed for this purpose, the TCCHI provides a theoretically grounded framework for measuring healthcare professionals’ integrated ability to utilize technological competency with ethical, person-centered care practices. This ensures that technology consistently serves to enhance patient dignity and holistic well-being within modern, team-based healthcare settings.

This study establishes the foundational content validity of the TCCHI; however, the next crucial step involves further psychometric validation. Future research will focus on large-scale data collection to conduct CFA to confirm the instrument’s factorial structure and assess its internal consistency and reliability across diverse professional populations.

## Figures and Tables

**Figure 1 healthcare-13-03003-f001:**
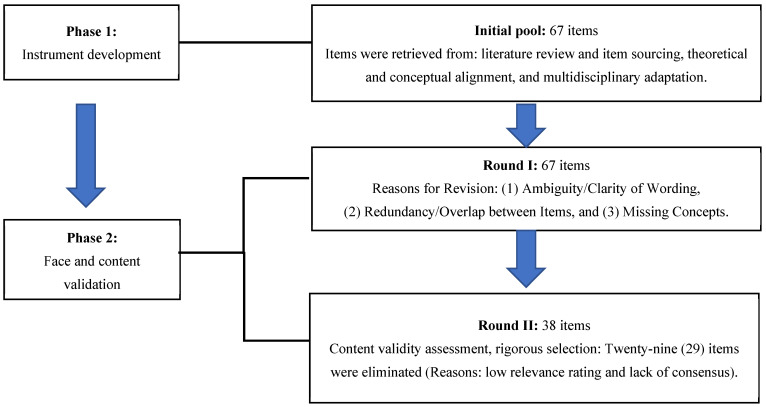
Item development process.

**Table 1 healthcare-13-03003-t001:** Characteristics of the expert panel.

Characteristics	Respondent (*n* = 10)
	Frequency (%)
Gender	
Male	5 (50)
Female	5 (50)
Age, years	
30–39	2 (20)
40–49	3 (30)
50–59	3 (30)
60–69	2 (20)
Working period in healthcare	
10–29 years	5 (50)
More than 30 years	5 (50)
Discipline	
Medicine	1 (10)
Nursing	7 (70)
Physical Therapy	2 (20)
Institutional affiliation/Occupation	
University hospital staff	1 (10)
University staff	7 (70)
JICA consultant	1 (10)
Director of nursing	1 (10)
Educational attainment	
MA	1 (10)
PhD	9 (90)

JICA: Japan International Cooperation Agency.

**Table 2 healthcare-13-03003-t002:** Developed TCCHI items.

Concept No.	No.	Questions	Med	Min–Max	I-CVI	UA	W	*p*
C 2	Q 7	Being close to the patient and respecting their rights.	9	6–9	**0.9**	1	12.5	0.70
C 4	Q 33	Focusing not only on the patient’s medical condition but also on the problems they face in daily life.	9	4–9	**0.9**	1	11.0	0.86
C 2	Q 1	Always treating every patient with compassion.	8	3–9	**0.9**	1	19.0	0.97
C 3	Q 13	Working with other professionals to support patients in realizing their dreams and hopes.	8	4–9	**0.9**	1	29.5	0.82
C 5	Q 42	Providing necessary and sufficient information to support patients and their families’ decision-making.	8	3–9	**0.9**	1	13.0	0.74
C 5	Q 45	Understanding and practicing what is best healthcare for patients.	8	6–9	**0.9**	1	22.5	0.76
C 6	Q 49	Always consider ethical issues that arise when providing care.	8	6–9	**0.9**	1	7.5	0.15
C 1	Q 66	Working in a way that allows you to grow as a healthcare professional.	8	2–9	**0.9**	1	27.0	0.91
C 4	Q 17	If the patient wishes, work with the patient’s family and supporters.	8	4–9	**0.9**	1	24.0	0.96
C 5	Q 22	Pay attention to the patient’s dreams, hopes, and requests, and support the patient’s self-actualization.	8	3–9	**0.9**	1	15.0	0.85
C 3	Q 24	Technology is useful for correctly assessing a patient’s condition.	8	5–9	**0.9**	1	34.0	0.93
C 5	Q 46	Striving to provide support that respects the patient’s self-determination.	8	3–9	**0.9**	1	20.5	0.67
C 6	Q 54	Contributing to creating a workplace where colleagues can freely exchange opinions.	8	5–9	**0.9**	1	17.5	0.76
C 1	Q 58	Providing healthcare services with a correct understanding of technology’s meaning and significance.	7	5–9	**0.9**	1	24.0	0.96
C 1	Q 59	Discussing the matter in a team conference in which the patient also participates, when worried about a patient’s complaint.	7	3–9	**0.9**	1	21.0	0.90
C 1	Q 60	Constantly updating knowledge of healthcare and welfare as a professional.	7	4–9	**0.9**	1	32.0	0.98
C 1	Q 65	Considering how to improve healthcare quality by reflecting on the care process with the patient.	7	6–9	**0.9**	1	14.0	0.97
C 1	Q 67	Communicating and sharing experiences gained through patient care with colleagues and medical/health science students.	7	2–9	**0.9**	1	33.5	0.99
C 2	Q 3	Building relationships that patients can trust.	7	3–9	**0.9**	1	20.5	0.89
C 2	Q 8	Speaking and acting in a way that earns the patient’s trust.	7	3–9	**0.9**	1	16.0	0.90
C 2	Q 9	Understanding the person not only based on their current illness and symptoms but also on their upbringing and lifestyle history.	7	3–9	**0.9**	1	11.5	0.20
C 2	Q 12	Focusing on the whole picture of the patient who needs care, not just the patient’s illness or disability.	7	3–9	**0.9**	1	29.0	0.95
C 5	Q 14	Developing a treatment plan with the patient.	7	1–9	**0.9**	1	14.5	0.57
C 3	Q 15	Appropriately reflecting the patient’s wishes in the treatment policy.	7	3–9	**0.9**	1	17.0	0.47
C 3	Q 18	Determining appropriate technology for patient care.	7	5–9	**0.9**	1	31.0	0.86
C 4	Q 29	Interacting appropriately according to the patient’s physical/psychological condition, which changes depending on the situation.	7	3–9	**0.9**	1	15.0	0.60
C 3	Q 23	Understanding the complete picture of the patient.	7	3–8	**0.9**	1	23.0	0.95
C 3	Q 25	Devising and implementing an appropriate care plan tailored to each patient’s individual needs.	7	3–9	**0.8**	0	21.0	0.69
C 6	Q 27	Providing medical treatment and care after understanding your own practical abilities.	7	3–8	**0.8**	0	17.5	0.50
C 4	Q 35	Providing care to ensure patients’ physical and mental comfort.	7	4–9	**0.8**	0	19.5	0.98
C 4	Q 36	Making a goal for patients to be able to live a life that is true to themselves.	7	5–9	**0.8**	0	36.0	1.00
C 5	Q 40	Requesting an explanation from the necessary professionals if the patient does not fully understand their condition or treatment.	7	6–9	**0.8**	0	50.0	0.99
C 4	Q 43	Supporting patients in improving their self-care abilities.	7	5–9	**0.8**	0	33.0	0.99
C 6	Q 48	Holding regular conferences with the healthcare team to minimize restrictions on patients’ behavior.	7	3–9	**0.8**	0	14.5	0.83
C 6	Q 56	Developing sensitivity to ethical issues.	7	1–9	**0.8**	0	13.0	0.47
C 2	Q 47	Recognizing the patient as an individual and irreplaceable person.	7	3–9	**0.8**	0	24.0	0.96
C 6	Q 53	Understanding the wishes of patients who are unable to express them and speaking on their behalf when necessary.	7	3–8	**0.8**	0	8.5	0.20
C 6	Q 55	Reporting inappropriate comments or behaviors toward patients by colleagues to management.	7	3–8	**0.8**	0	25.5	0.98
C 1	Q 57	Always learning to adapt to new technology.	6	4–9	0.7	0	34.0	0.99
C 1	Q 62	Striving to grow together in medical situations shared by healthcare professionals, patients, and families.	6	3–9	0.6	0	25.5	0.87
C 1	Q 64	Improving yourself to become familiar with the latest medical equipment in your department.	6	3–9	0.6	0	28.0	0.99
C 2	Q 5	Using technology to understand patient deeply.	6	3–9	0.6	0	22.0	0.93
C 2	Q 6	Sharing necessary information with the patient in order to understand them.	6	2–9	0.6	0	24.5	0.97
C 2	Q 10	Evaluating the stress and anxiety that arise for both parties in the medical professional-patient relationship.	6	2–8	0.6	0	33.0	0.99
C 3	Q 19	Accepting and respecting the changing wishes of patients.	6	4–9	0.6	0	28.5	0.94
C 3	Q 21	Respecting the patient’s wishes and supporting them with a focus on their recovery.	6	3–8	0.6	0	36.0	0.95
C 3	Q 30	Using caring competence to understand patients and their families.	6	2–8	0.6	0	17.5	0.75
C 3	Q 32	Providing flexible patient care according to time and circumstances.	6	3–8	0.5	0	25.0	0.86
C 5	Q 41	Providing the patient with the necessary and sufficient information to make his/her own decisions, then explaining and obtaining consent.	6	2–9	0.4	0	27.0	0.91
C 6	Q 50	Respecting patient privacy and observing patients as necessary to ensure patient safety.	6	2–9	0.3	0	18.0	0.53
C 6	Q 51	Listening to patients with compassion.	6	3–9	0.3	0	27.5	0.92
C 6	Q 52	Improving patient experience by correctly understanding and using technology to eliminate constraints and restrictions.	6	3–9	0.3	0	28.0	0.93
C 1	Q 63	Using technological competency to know the patient comprehensively.	6	3–8	0.2	0	20.0	0.98
C 2	Q 2	Empathizing with the patient’s experiences and emotions.	6	3–9	0.1	0	21.0	0.90
C 2	Q 4	Helping patients express their hopes and dreams.	6	3–9	0.1	0	38.0	0.97
C 3	Q 20	Providing care according to the patient’s health condition.	6	3–9	0.1	0	32.5	0.98
C 4	Q 37	Focusing on providing mental care to help patients live peaceful lives.	6	3–8	0	0	25.0	0.98
C 1	Q 61	Considering the use of technology from the perspective of caring.	5	3–9	0	0	46.5	0.98
C 2	Q 11	Striving to understand the patient’s personality and developmental characteristics.	5	1–9	0	0	44.5	0.96
C 3	Q 28	Healthcare professionals coordinate with other professionals to meet patients’ needs.	5	1–8	0	0	34.0	0.99
C 4	Q 34	Improving patients’ quality of life as a treatment goal.	5	3–7	0	0	43.0	0.99
C 3	Q 16	Actively collaborating with other professionals to fulfill the patient’s wishes.	4	3–7	0	0	55.0	1.00
C 3	Q 26	Communicate actively with patients and develop care plans based on mutual understanding to provide high-quality medical services.	4	1–8	0	0	41.5	0.99
C 5	Q 39	Helping patients live a quality of life.	4	2–8	0	0	10.0	0.79
C 5	Q 44	Sharing what you have noticed during your interactions with patients.	4	3–9	0	0	27.5	0.92
C 3	Q 31	Providing care and treatment with the utmost consideration for each patient, regardless of physical function.	3	1–9	0	0	28.0	0.99
C 4	Q 38	Enhancing patient self-esteem through technology-enabled care.	3	1–8	0	0	32.5	0.98

Note: UA (Universal agreement) total = 27, S-CVI/Ave = 0.55, S-CVI/UA = 0.02. The average proportion of items judged to be relevant by the 10 experts was 0.66. C1: promoting self-growth and technological learning; C2: building trusting relationships with patients; C3: providing person-centered care through the appropriate use of technology; C4: enhancing the physical and emotional comfort of patients; C5: promoting patient learning and growth; C6: engaging in ethico-moral practice regarding technology use and patient advocacy. W = Wilcoxon matched-pairs signed-rank test. The ICC showed moderate: Round, ICC = 0.486 (95% CI, 0.291–0.648) and Round 2, ICC = 0.501 (95% CI, 0.317–0.655).

## Data Availability

The data presented in this study are available upon request from the corresponding author. These data are not publicly available due to privacy restrictions.

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
