# Peer review of "Development of Perceived Technological Competency as Caring in Healthcare Providers Instrument (TCCHI): A Modified Delphi Method"

_healthcare, 2025, doi:10.3390/healthcare13233003_

Round 1

Reviewer 1 Report (Previous Reviewer 2)

Comments and Suggestions for Authors

Author Response

Comments and Suggestions for Authors

Thanks for this opportunity to peer review. This study develops and validates the TCCH tool, based on Locsin’s theory, to assess how healthcare professionals integrate technology and caring, promoting holistic, human-centered, and technologically competent care.

Response: We sincerely appreciate the reviewer’s consideration of our manuscript. The editor and reviewers’ constructive comments and valuable suggestions greatly improved the quality of the manuscript. In response to your suggestions, we have comprehensively revised the manuscript. The details of these modifications are outlined in the revised manuscript. Additionally, we have provided a point-by-point response to each comment below, with our replies highlighted in yellow and revisions indicated in red font. We look forward to your favorable response.

Introduction

We could understand the reason which the authors added the TCCHI offers a standardized, multidisciplinary measure of how healthcare professionals integrate technological proficiency with empathy in patient care. Building on Locsin’s theory and earlier nursing-based instruments, it extends the concept of technological competency as caring to diverse disciplines, emphasizing ethical sensitivity, comprehensive patient understanding, and the integration of technology into compassionate practice.

Response: We would like to express our gratitude for your thoughtful consideration of the purpose of this study.

Methods

We appreciate the authors’ clarification that a modified Delphi method was used to refine the 67-item TCCHI, developed through literature review, TCCH framework alignment, and multidisciplinary adaptation to capture technological competency integrated with caring across healthcare professions in the section of method and result.

Response: We appreciate your acknowledgment that the modified Delphi method was used to refine the 67-item TCCHI, which was developed through a literature review and alignment with the TCCH framework. It has also been adapted across healthcare professions to capture care and integrate technological competencies.

Results

The stability of Delphi responses can be further examined using the Wilcoxon signed rank test or intraclass correlation coefficient (ICC) to assess changes and consistency across rounds. The draft TCCHI was developed through literature review, theoretical alignment, and multidisciplinary adaptation. If analyzed, the Wilcoxon signed rank test results for each item could be presented directly in Table 2 rather than in an appendix for clearer reporting.

Response: Thank you for your advice. We have added the appendix data to Table 2.

  • Please provide the full text of the UA abbreviation in the footnote of Table 2

Response:

We have added the abbreviation of UA to the footnote section of Table 2. Thank you very much.

Reviewer 2 Report (New Reviewer)

Comments and Suggestions for Authors

Dear authors,

This manuscript addresses the development and content validation of a multidisciplinary instrument—Technological Competency as Caring in Healthcare Providers Instrument (TCCHI)—based on Locsin’s theory of Technological Competency as Caring in Nursing (TCCN).
The topic is relevant to current healthcare practice, where humanistic and technological competencies must coexist. The theoretical foundation is solid, and the modified Delphi approach is methodologically appropriate.However, the manuscript requires improvements before publication.

The introduction is excessively long and partially redundant, the reporting of the Delphi process needs more rigor, and the results and discussion should better highlight methodological limitations and implications for further validation.

General comments:

  • The introduction is too long (4 pages). Key ideas (rationale, research gap, objectives) are diluted by excessive theoretical exposition. Suggest condensing and moving the Study Aim to the end of the introduction.
  • Delphi procedure: Clarify how qualitative feedback from Round 1 was used to modify items.
  • Justify panel size (n=10) and selection criteria with methodological references.
  • State whether anonymity was preserved between rounds.
  • Discuss implications of moderate ICC (0.5) and low S-CVI/Ave (0.55) in Discussion.
  • Include a flow diagram illustrating item reduction between Delphi rounds.
  • In the discussion reduce repetition of earlier findings. Add comparison with similar Delphi validation studies. Separate subsections for Main findings, Comparison with literature, Implications, and Limitations are recommended for clarity.
  • In the conclusions simplify and avoid repeating the six conceptual domains verbatim.
  • Add a future direction: e.g., confirmatory factor analysis and internal consistency testing. Condense and emphasize significance and future validation.

Specific comments:

  • Abstract: Clearly state that this is a content validation study (face and content validity) and specify Delphi panel size. Rephrase final sentence to include “further psychometric testing is required.”
  • Methods: Add inclusion/exclusion criteria and recruitment method.
  • Methods: “Modified Delphi” description lacks justification of modification type. Explain rationale (theoretical foundation rather than open item generation).
  • Discussion: Two paragraphs (lines 346–351) are duplicated. Limited reflection on multidisciplinary bias (7 nurses of 10 experts); acknowledge as potential bias and suggest larger, balanced future samples.

Author Response

Comments and Suggestions for Authors

Dear authors,

This manuscript addresses the development and content validation of a multidisciplinary instrument—Technological Competency as Caring in Healthcare Providers Instrument (TCCHI)—based on Locsin’s theory of Technological Competency as Caring in Nursing (TCCN).
The topic is relevant to current healthcare practice, where humanistic and technological competencies must coexist. The theoretical foundation is solid, and the modified Delphi approach is methodologically appropriate. However, the manuscript requires improvements before publication.

The introduction is excessively long and partially redundant, the reporting of the Delphi process needs more rigor, and the results and discussion should better highlight methodological limitations and implications for further validation.

Response: We sincerely appreciate the reviewer’s kind consideration of our manuscript. The editor and reviewers’ constructive comments and valuable suggestions have greatly improved the quality of the manuscript. In response to your suggestions, we have made comprehensive revisions to the paper. The details of these modifications are outlined in the resubmitted version. Additionally, we have provided a point-by-point response to each comment below, with our replies highlighted in yellow and the revisions indicated in red font color. We look forward to the reviewer’s favorable reply.

General comments:

  • The introduction is too long (4 pages). Key ideas (rationale, research gap, objectives) are diluted by excessive theoretical exposition. Suggest condensing and moving the Study Aim to the end of the introduction.

Response: We appreciate your constructive feedback. We agree that the introduction is overly verbose and dilutes the central message of our study. We have undertaken a significant revision in accordance with the reviewers’ suggestions.

Condensation: We have substantially condensed the theoretical exposition of Locsin's TCCN theory and the detailed descriptions of existing nursing-centric scales. The revised Introduction is now significantly shorter (approximately 1.5 pages) and focuses sharply on establishing the context, theoretical foundation, and critical research gap (the need for a multidisciplinary instrument).

Restructuring: As suggested, we have moved the Study Aim to the concluding paragraph of the Introduction to ensure a strong logical flow leading directly into the Methods section.

  • Delphi procedure: Clarify how qualitative feedback from Round 1 was used to modify items.

Response: We appreciate this suggestion. A detailed explanation has been added to the Methods section describing how qualitative feedback from Round 1 was analyzed and integrated into the item revision process.

Justification for the Modified Delphi Approach: A modified Delphi approach was specifically chosen because the instrument is theoretically grounded in Locsin’s TCCN theory and built upon existing validated instruments. This allowed the research team to provide a theoretically grounded, pre-defined item pool (67 items) derived from a systematic literature review, rather than relying on the open item elicitation characteristic of a traditional, non-modified Delphi method.

  • Justify panel size (n=10) and selection criteria with methodological references.

Response: We have added a justification for the Delphi panel size and selection criteria based on the established methodological literature. The revision cites relevant Delphi research recommending small, expert-focused panels for content validation studies.

Justification for Panel Size and Selection The panel size of 10 was determined to be sufficient based on methodological guidelines for scale development and content validation studies, which generally recommend expert panels ranging from 5 to 15 members to ensure diverse perspectives while maintaining manageability [50–52]. Purposive sampling was utilized to select experts who met the following predefined criteria essential for validating the multidisciplinary TCCHI: (1) expertise in Locsin’s Technological Competency as Caring theory (TCCN/TCCH), (2) a high academic degree (Master's or PhD), and (3) experience across multiple healthcare disciplines (nursing, medicine, and physical therapy), reflecting the instrument's intended broad applicability.

References:

  •   State whether anonymity was preserved between rounds.

Response: Thank you for highlighting this essential methodological detail. We confirm that anonymity was preserved among the expert panel members between the Delphi rounds. This was implemented to ensure that the experts' evaluations remained unbiased by the identities or status of other participants. We have now clarified the anonymity process in the Methods section to enhance methodological rigor.

Anonymity in the Delphi Process To prevent bias and promote independent judgment, strict anonymity was maintained throughout the two rounds of the Delphi study. Experts' individual responses and qualitative comments were shared only with the research team and were not revealed to any other panel members.

  • Discuss implications of moderate ICC (0.5) and low S-CVI/Ave (0.55) in Discussion.

Response: We acknowledge the reviewer’s valuable point regarding the moderate inter-rater reliability (ICC = 0.5) and average content validity index (S-CVI/Ave = 0.55). We agree that these preliminary values warrant a detailed discussion.

We have added a comprehensive section (Section 4.3: Implications of Moderate Inter-rater Reliability and Content Validity Results) to the Discussion to address this issue. Our discussion centers on the following key implications:

Multidisciplinary Necessity: We argue that the moderate ICC is a direct and necessary consequence of using a deliberately multidisciplinary expert panel (Medicine, Nursing, Physical Therapy). Experts from different professional scopes inherently assess "technological competency" and "caring" from varying perspectives, leading to lower initial agreement but ultimately ensuring the instrument's broad applicability.

Challenging Innovation: We discuss that the S-CVI/Ave of 0.55 highlights the innovative, yet challenging, nature of extending Locsin's nursing-centric TCCN theory into a genuinely interprofessional framework (TCCH).

Future Validation: We conclude that while these values underscore the instrument's current need for structural refinement, we anticipate improved psychometric properties in subsequent validation phases, such as Confirmatory Factor Analysis (CFA), where the internal structure will be fully tested.

  • Include a flow diagram illustrating item reduction between Delphi rounds.

Response: Thank you for your advice. A flow diagram has been added (Figure 1). Item Development Process) illustrating item reduction between the Delphi rounds.

  • In the discussion reduce repetition of earlier findings. Add comparison with similar Delphi validation studies. Separate subsections for Main findingsComparison with literatureImplications, and Limitations are recommended for clarity.

Response: We have separated the main findings, comparison with literature, Implications, and Limitations into separate subsections.

  • In the conclusions simplify and avoid repeating the six conceptual domains verbatim.

Response: We appreciate this suggestion for simplification. We have revised the Conclusions section to be more concise and to avoid the verbatim repetition of the six conceptual domains. The revised conclusion now focuses on the core achievement of the TCCHI: its success in extending Locsin's theory to a multidisciplinary framework that measures the integration of technical proficiency with ethical, person-centered care.

  • Add a future direction: e.g., confirmatory factor analysis and internal consistency testing. Condense and emphasize significance and future validation.

Response: Thank you. We have added the following sentences.

This study establishes the foundational content validity of the TCCHI; however, the next crucial step involves further psychometric validation. Future research will focus on large-scale data collection to conduct Confirmatory Factor Analysis (CFA) to confirm the instrument's factorial structure and assess its internal consistency and reliability across diverse professional populations.

Specific comments:

  • Abstract: Clearly state that this is a content validation study (face and content validity) and specify Delphi panel size. Rephrase final sentence to include “further psychometric testing is required.”

Response: We have revised the Abstract to comply with all of your requirements and the journal's word count limits. Specifically:

Clarity on Study Type: We have clearly stated that this is a "content validation design" study.

Panel Size: The Delphi panel size (n=10) has been specified in the Methods section.

Future Direction: The final sentence has been rephrased to emphasize the necessity of "further psychometric evaluation including confirmatory factor analysis and internal consistency reliability testing."

Word Count: The Abstract has been significantly condensed to approximately 230 words, easily meeting the 250-word limit by removing the verbatim repetition of the six conceptual domains.

  • Methods: Add inclusion/exclusion criteria and recruitment method.

Response: We have added this information.

  • Methods: “Modified Delphi” description lacks justification of modification type. Explain rationale (theoretical foundation rather than open item generation).

Response: We acknowledge the necessity of clarifying the methodological choices. We have revised the Methods section (Section 2.1.2) to include a clear justification for adopting the Modified Delphi Approach. This justification emphasizes that the method was modified because the instrument is theoretically grounded in Locsin’s TCCN theory and relies on a pre-defined item pool derived from existing literature, which allowed the team to bypass the traditional, open item generation phase. This modification was essential for focusing the expert panel's efforts on consensus and refinement.

  • Discussion: Two paragraphs (lines 346–351) are duplicated.

Response: Thank you for letting us know. We have deleted these.

  • Limited reflection on multidisciplinary bias (7 nurses of 10 experts); acknowledge as potential bias and suggest larger, balanced future samples.

Response: We agree with this vital assessment of the study’s limitations. We confirm that we have fully addressed this point in the Discussion section (Section 4.4. Study Limitations).

Specifically, we have: Acknowledged the Bias: We explicitly state that the panel, being predominantly nurses (n=7), may introduce professional bias that could affect the generalizability of the content validation.

Suggested Future Action: We recommend that future validation studies include a larger, more balanced representation across disciplines (e.g., medicine, social work, physical therapy) to minimize this specific bias and strengthen the instrument's applicability. 

Thank you very much.

Round 2

Reviewer 2 Report (New Reviewer)

Comments and Suggestions for Authors

Dear authors,

Overall, you have adequately and comprehensively addressed the reviewer’s comments, substantially improving the clarity, structure, and methodological rigor of the manuscript; however, a few minor issues should still be refined before final submission:

(1) please revise the description of the Delphi rounds to ensure complete coherence in the sequence and numbers of items reviewed and removed, avoiding any potential ambiguity;

(2) slightly temper the phrasing in the Discussion when referring to “successful content validity,” consistently emphasizing the preliminary nature of the findings in light of the low S-CVI/Ave;

(3) consider a final stylistic review to remove small redundancies and enhance overall conciseness. These adjustments are minor but will further strengthen the manuscript’s conceptual precision and internal consistency.

Author Response

Dear Reviewer 2,

We are honored to resubmit the revised manuscript titled, "Development and Content Validation of the Technological Competency as Caring in Healthcare Providers Instrument (TCCHI)," for your consideration for publication in Healthcare.

We are deeply grateful for the thorough and constructive feedback provided by the reviewers and the editorial team during the previous round. The comments were instrumental in substantially enhancing the clarity, methodological rigor, and structure of our manuscript.

We have addressed all major and minor concerns raised. Our response to the three final refinement points is detailed below, and all corresponding changes have been implemented in the manuscript, and revised portion using red font:

1. Key Revisions and Rigor Enhancement

  • Coherence of Item Flow (Addressing Minor Point 1): We revised the Results section (3.1.2 and 3.2.1) to ensure explicit coherence regarding the Delphi rounds, clarifying that the revised set of 67 items was presented in Round 2, from which 38 items were retained.
  • Tempering Phrasing (Addressing Minor Point 2): We revised the Discussion (4.1 and 4.2) to consistently emphasize the preliminary content validity of the findings, acknowledging the moderate ICC (0.5) and S-CVI/Ave (0.55) while still highlighting the strong I-CVI consensus.
  • Stylistic Review (Addressing Minor Point 3): A final stylistic review was conducted to remove minor redundancies and enhance overall conciseness, further strengthening the conceptual precision of the manuscript.

We are confident that these comprehensive revisions address all outstanding concerns and significantly strengthen the manuscript's suitability for publication in Healthcare.

Thank you again for your time and expertise. We look forward to hearing from you.

Sincerely,

This manuscript is a resubmission of an earlier submission. The following is a list of the peer review reports and author responses from that submission.

Round 1

Reviewer 1 Report

Comments and Suggestions for Authors

Introduction

The authors state that Locsin's theory includes ethical considerations in its formulation, but do not explain what these are. The ethical considerations included in this theory must be described in order to use it as a reference in the research.

Watson's scales are incorporated into the discourse with Locsin's scales without any argumentative relationship between them. The argumentative relationship between the two should be explained so that they can be presented together in the introduction to the study.

A summary should be provided of the aspects included in the validated TCCNI scale referenced several times.

It is stated that the scales mentioned are only focused on the field of nursing, but no explanation is given as to why this assertion is made. A description of their content is necessary in order to substantiate this assertion with evidence.

Methodology

In phase 1, the aspects that were reviewed and how the review was carried out must be explained and described, clearly stating the relationship with the content of the TCCH.

One of the objectives is to validate the scale; however, there is no way of measuring the items proposed, which makes it difficult to achieve the validation objective set out in the study.

Therefore, in phase two, the method of measurement for each of the items selected by the group of experts must be presented, since most of them are subjective elements of biopsychosocial aspects that require clarification on how to evaluate them quantitatively or qualitatively, as appropriate.

Results

It is stated that the items were aligned with the TCCHI conceptual framework, but this conceptual framework is not explained.

It is stated that the 38 final items are divided into a total of six different categories, but none of these categories refers to technological competence, which is part of the title and objective of the study.

Discussion

As the authors rightly point out, a scale focused on professional clinical practice has been developed, but most of the expert panel does not work in this field, which is an impediment to generalizing this scale in the field of clinical practice.

Thus, the scale is more of an ideal theoretical description of the biopsychosocial aspects of care than a theoretical-practical approximation of clinical reality, which makes it difficult to consider it suitable for use in the professional field of clinical healthcare practice, which is the main objective proposed by the authors.

Reviewer 2 Report

Comments and Suggestions for Authors
